# Contrastive Self-Supervised Learning As Neural Manifold Packing

**Guanming Zhang**[1,2]  **David J. Heeger**[2,3]  **Stefano Martiniani**[1,2,4,5]

[1] Center for Soft Matter Research, Department of Physics, New York University
[2] Center for Neural Science, New York University
[3] Department of Psychology, New York University
[4] Simons Center for Computational Physical Chemistry, New York University
[5] Courant Institute of Mathematical Sciences, New York University

`{gz2241, david.heeger, sm7683}@nyu.edu`

## Abstract

Contrastive self-supervised learning based on point-wise comparisons has been widely studied for vision tasks. In the visual cortex of the brain, neuronal responses to distinct stimulus classes are organized into geometric structures known as neural manifolds. Accurate classification of stimuli can be achieved by effectively separating these manifolds, akin to solving a packing problem. We introduce **C**ontrastive **L**earning **A**s **M**anifold **P**acking (CLAMP), a self-supervised framework that recasts representation learning as a manifold packing problem. CLAMP introduces a loss function inspired by the potential energy of short-range repulsive particle systems, such as those encountered in the physics of simple liquids and jammed packings. In this framework, each class consists of sub-manifolds embedding multiple augmented views of a single image. The sizes and positions of the sub-manifolds are dynamically optimized by following the gradient of a packing loss. This approach yields interpretable dynamics in the embedding space that parallel jamming physics, and introduces geometrically meaningful hyperparameters within the loss function. Under the standard linear evaluation protocol, which freezes the backbone and trains only a linear classifier, CLAMP achieves competitive performance with state-of-the-art self-supervised models. Furthermore, our analysis reveals that neural manifolds corresponding to different categories emerge naturally and are effectively separated in the learned representation space, highlighting the potential of CLAMP to bridge insights from physics, neural science, and machine learning.

## 1 Introduction

Learning image representations that are both robust and broadly transferable remains a major challenge in computer vision, with critical implications for the efficiency and accuracy of training across downstream tasks. State-of-the-art models address this problem by employing contrastive self-supervised learning (SSL), wherein embeddings are treated as vectors and optimized using pairwise losses to draw together positive samples, augmented views of the same image, and push apart negative samples, views of different images [1, 2, 3, 4, 5, 6]. While SSL has advanced significantly, and in some cases, surpassed supervised methods [7, 8], its geometric foundations remain largely underexplored. In visual cortex, neural activity often resides within geometric structures known as neural manifolds — which span the embedding space [9] and may exhibit low intrinsic dimensionality [10] that capture much of the observed variability in neural activity. Neural manifolds have been characterized in a number of neural systems supporting a variety of cognitive processes

39th Conference on Neural Information Processing Systems (NeurIPS 2025).

(in addition to vision) including neural activity coordination, motor control, and information coding [11, 12, 13, 14]. In the context of SSL, neural representations of similar and dissimilar samples are frequently conceptualized as distinct sub-manifolds in the representation space, where a *sub-manifold* corresponds to the encoding of a single data sample and its transformations, and a *manifold* to an entire class. In classification, the objective is to separate these class-level manifolds so that each maps cleanly to a distinct category under a readout function. When manifolds corresponding to different classes, for example, manifolds representing "cats" and "dogs" overlap in neural state space, the resulting ambiguity can lead to misclassification. This viewpoint recasts classification as the problem of efficiently arranging manifolds in representation space to minimize overlap and enhance separability [15].

Our main contributions are as follows:

- We propose a loss function inspired by the interaction energy of short-range repulsive particle systems such as simple liquids and jammed packings, endowing each hyperparameter with a clear physical interpretation.

- We demonstrate that CLAMP achieves state-of-the-art image classification accuracy, matching state-of-the-art SSL methods and even setting a new SOTA on ImageNet-100, under the standard linear evaluation protocol (a frozen backbone plus a linear readout), all while being remarkably simple and interpretable thanks to its single projection head and one-term, physics-grounded loss comprising a single term.

- We show that the dynamics of SSL pretraining under CLAMP resemble those of the corresponding interacting particle systems, revealing a connection between SSL and non-equilibrium physics.

- We compare CLAMP's internal representations to mammalian visual cortex recordings and find that CLAMP offers a compelling account of how complex visual features may self-organize in the cortex without explicit supervision.

## 2 Related work

### 2.1 Interacting particle models in physics

A particularly insightful analogy for CLAMP comes from random organization models, originally introduced to describe the dynamics of driven colloidal suspensions [16, 17, 18]. In these models, overlapping particles are randomly "kicked" while isolated particles remain static. As the volume fraction is increased, the system undergoes a phase transition from an absorbing state (where no particles overlap) to an active steady state (where overlaps cannot be resolved under the prescribed dynamics) in two and three dimensions [19]. When kicks between overlapping particle pairs are reciprocal, meaning that each particle is displaced by an equal but opposite amount, the critical and active states are characterized by anomalously suppressed density fluctuations at large scales, a property known as "hyperuniformity"[20] or "blue noise"[21, 22]. This absorbing-to-active transition coincides with random close packing (RCP) when the noisy dynamics are reciprocal [17]. Recent work further establishes an equivalence between this process and a stochastic update scheme that minimizes a repulsive energy under multiplicative noise [15]. We note that even beyond the hyperuniformity observed for dense active states of random organization systems, for a high-dimensional system of short-range, repulsive particles, such as those interacting via an exponential potential ($e^{-cr}, c > 0$), confined to a closed surface, a uniform spatial distribution minimizes the system's energy in the large density limit [23, 24, 25]. Drawing on these insights, CLAMP employs a short-range repulsive potential over augmentation sub-manifolds as its self-supervised learning loss, favoring the separation of embeddings.

### 2.2 Feature-level contrastive learning

Following recent SOTA self-supervised learning models, we use image augmentations and contrastive losses. Most self-supervised learning frameworks treat features as points or vectors in the embedding space, contrasting positive samples—embeddings of augmented views of the same image—with negative samples. They typically use a pairwise contrastive loss such as InfoNCE [1, 2, 3, 4, 5, 6], or a weighted-sample InfoNCE loss where negative samples are drawn from tailored Boltzmann-like

distributions to sharpen sample-wise contrasts [26]. Here, CLAMP treats each image as its own class; however, instead of applying the contrastive loss directly to individual feature vectors, it operates on the entire augmentation sub-manifolds surrounding each input embedding.

## 2.3 Clustering methods for self-supervised learning

Clustering-based self-supervised learning methods assign images to clusters based on similarity criteria, rather than treating each image as a distinct class. For instance, The DeepCluster [27] applies the k-means algorithm to assign pseudo-labels to embeddings, which serve as training targets in subsequent epochs. Collapse-resistant non-contrastive methods prevent feature collapse by using a fixed dictionary and enforcing augmentation invariance without negative samples [28]. However, the DeepCluster approach requires k-means classification on the entire dataset, making it unsuitable for online, mini-batch, learning. SwAV [29] overcomes this limitation by using soft assignments and jointly learning cluster prototypes and embeddings, enabling flexible online updates. In contrast to clustering methods, our approach groups each image with its augmentations to learn the geometry of the augmentation sub-manifold without clustering across different samples.

## 2.4 Alignment-Uniformity and maximum manifold capacity methods

The Alignment-Uniformity method [30] constructs the loss function by incorporating two components: an alignment term and a uniformity term. The alignment term maximizes the cosine similarity of the embedding vectors between positive pairs, thereby encouraging consistency. The uniformity term acts as a repulsive pairwise potential over all pairs of embeddings, driving the representations to be uniformly distributed on the hyper-sphere and mitigating representational collapse [31].

Maximum manifold capacity theory measures the linear binary classification capacity of manifolds with random binary labels. For $P$ $d$-dimensional manifolds embedded in $D$ dimensions, $\alpha_c$ measures the maximum capacity such that for all $P/D < \alpha_c$, these manifolds can be linearly classified into two categories with probability 1 in the limit where $P \to \infty$, $D \to \infty$, $d/D \to 0$, and $P/D$ remains finite [32, 33]. Maximum manifold capacity representation (MMCR) maximizes this capacity on augmentation sub-manifolds in the framework of multi-view self-supervised learning and achieves high accuracy classification performance [34].

Unlike MMCR, which expressly maximizes the representation's *binary linear* classification capacity, CLAMP optimizes the embeddings for an *n-ary nonlinear* classification problem, making the two approaches fundamentally different. Moreover, CLAMP's loss function is based on short-range interactions: the contribution from each augmentation sub-manifold depends only on its neighboring sub-manifolds, whereas MMCR requires global information from all sub-manifolds. Finally, although MMCR has comparable computational complexity to CLAMP, MMCR is less efficient to train, because it requires one SVD for each learning step, compared to CLAMP, and its loss function and optimization dynamics lack the clear, physics-based interpretability that CLAMP provides.

Compared to the Alignment-Uniformity method, our approach differs in three key aspects: (1) The Alignment-Uniformity loss is defined over feature vectors, whereas our loss operates directly on manifolds; (2) Our method employs a single loss term that simultaneously enforces alignment and uniformity, avoiding the need to manually balance separate terms as required in Alignment-Uniformity; (3) Our pairwise repulsion is short-ranged, involving only neighboring embeddings, which reduces computational complexity compared to the global (long-ranged) repulsion used in Alignment-Uniformity.

To summarize, both the MMCR and Alignment-Uniformity methods enhance mutual information between positive pairs by pulling together augmentations of the same image (positive samples) and pushing apart those of different images (negative samples) [35, 30]. In CLAMP, these properties emerge naturally from the short-range repulsive dynamics described in Sec. 2.1, without relying on global information or separate alignment and uniformity terms that need to be balanced, making CLAMP a particularly elegant, efficient, and interpretable solution.

# 3 Method

## 3.1 Description

Inspired by physical packing problems and recent advances in self-supervised learning, CLAMP learns structured representations by minimizing an energy-based loss that encourages optimal packing of neural manifolds in embedding space. Our framework is designed specifically for vision tasks that use images as input. As shown in Fig. 1, given a batch of images, denoted $x_1, x_2, \ldots, x_b$, where $b$ represents the batch size, we adopt the standard contrastive learning setup, in which positive samples are generated through image augmentations. For each input image $x_i$, we apply random transformations to obtain $m$ augmented views, denoted as $x_i^1, x_i^2, \ldots, x_i^m$.

To establish precise terminology, we define an augmentation sub-manifold as the set of embeddings produced from a finite number of augmentations of a single image. In the limit of infinite augmentations, this becomes a sub-manifold. The union of sub-manifolds corresponding to all images in a given class defines a class manifold, or simply a neural manifold.

We employ a backbone encoder $f_\theta(x)$ as the feature extractor to get the representations, $y_i^k = f_\theta(x_i^k)$. To map these representations into the embedding space, we apply a multilayer perceptron (MLP) projection head, $z_i^k = g_\theta(y_i^k)$. Normalizing embedding vectors on the unit hypersphere has been shown to be effective in self-supervised learning and variational autoencoders [34, 30, 36]. Therefore, we normalize the embeddings to lie on the unit hypersphere by subtracting the global center and projecting to unit norm, $\tilde{z}_i^k = (z_i^k - c)/\|z_i^k - c\|_2$, where $c = \frac{1}{mb}\sum_i \sum_k z_i^k$ is the global mean of all the projected vectors in the batch.

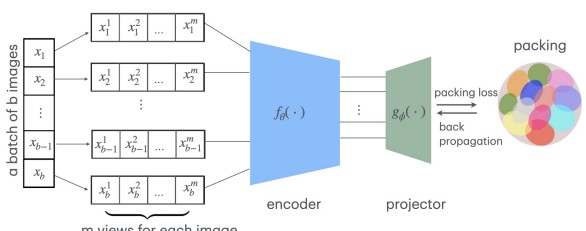

Figure 1: **CLAMP architecture**. The CLAMP framework processes a batch of $b$ input images by applying augmentations to generate $m$ views for each image. These augmented views are then encoded and projected into a shared embedding space. Within this space, the augmented embeddings corresponding to each input form a distinct sub-manifold, resulting in $b$ such sub-manifolds. Then, a pairwise packing loss is applied to minimize overlap between these sub-manifolds. The gradient of the loss is subsequently backpropagated to optimize the model.

For each image $x_i$, its $m$ augmentations form a cluster of projected vectors that we refer to as an augmentation sub-manifold, $\{\tilde{z}_i^1, \tilde{z}_i^2, \ldots \tilde{z}_i^m\}$ centered at $Z_i = \frac{1}{m}\sum_k \tilde{z}_i^k$. We approximate the shape of each augmentation sub-manifold with a high-dimensional ellipsoid, as described next in Sec. 3.1.1. This procedure yields $b$ ellipsoidal sub-manifolds, one per image. We treat sub-manifolds from different images as repelling entities and define the loss as $\mathcal{L} = \log(\mathcal{L}_{overlap})$, where $\mathcal{L}_{overlap} = \sum_{i,j} E(Z_i, Z_j)$ penalizes overlap between sub-manifolds. $E(Z_i, Z_j)$ is an energy-based pairwise repulsive potential that pushes ellipsoids $i$ and $j$ away from one another if they overlap,

$$\mathcal{L}_{overlap} = \begin{cases} \sum_{i \neq j} \left(1 - \frac{\|Z_i - Z_j\|_2}{r_i + r_j}\right)^2, & \text{if } \|Z_i - Z_j\|_2 < r_i + r_j \\ 0, & \text{otherwise} \end{cases} \quad, \text{where } r_i = r_s\sqrt{\frac{\text{Tr}(\Lambda_i)}{m}}. \quad (1)$$

Note that even though the loss function acts locally on neighboring sub-manifolds in the embedding space, each weight update affects all embeddings. Since the logarithm is a monotonic function, applying it to $\mathcal{L}_{overlap}$ preserves the locations of its minima while compressing large loss values, stabilizing the training process.

Following the convention established in [2], we refer to the output space of the encoder $f_\theta(\cdot)$ as the *representation space*, while the output space of the projection head $g_\phi(f_\theta(\cdot))$ is designated as the *embedding space*.

### 3.1.1 Approximate augmentation sub-manifolds as ellipsoids

We approximate the distribution of embedding vectors $p(\tilde{z}_i)$ generated from random augmentations of input image $x_i$ as a multivariate Gaussian with mean $\tilde{Z}_i$ and covariance $\Lambda_i$, $\tilde{z}_i \sim \mathcal{N}(Z_i, \Lambda_i)$. We assume that $\Lambda_i$ is full-rank and invertible (see Appendix K for a treatment of the general case). To represent the augmentation sub-manifold in embedding space, we define the corresponding enclosing

ellipsoid with Mahalanobis distance $r_s$,

$$(\tilde{z} - Z_i)\Lambda_i^{-1}(\tilde{z} - Z_i)^T = r_s^2 \qquad (2)$$

where $r_s$ is a scaling hyperparameter that controls the effective size of the manifold. A large $r_s$ corresponds to larger ellipsoids, and an appropriate choice of $r_s$ ensures that the ellipsoid encloses the majority of the augmentation sub-manifold (Fig. 2(a)). The lengths of the ellipsoid's semi-axes are given by the square roots of the nonzero eigenvalues of $\Lambda_i$ scaled by $r_s$: $r_s\sqrt{\lambda_i^{(1)}}, r_s\sqrt{\lambda_i^{(2)}}, ...$ where $\lambda_i^{(k)}$ is the k-th eigenvalue of $\Lambda_i$. To estimate the effective size of the ellipsoid, we compute the square root of the expected squared semi-axial length, $\sqrt{\mathbb{E}[(r_s\sqrt{\lambda})^2]}$, yielding the sub-manifold radius, $r_i = r_s\sqrt{\frac{\sum_j \lambda_i^{(j)}}{\text{rank}(\Lambda_i)}}$. The radius $r_i$ provides an upper bound on the volume of the ellipsoid $i$ ($V_i$), $r_i \geq cV_i^{1/\min(b,D)}$ (see Appendix K for details). In practice, the number of augmentations $m \sim O(10)$ is much smaller than the embedding dimension $D \sim O(100)$, making the empirical covariance $\Lambda_i$ lower rank or singular. In such cases, the expression for $r_i$ remains valid with $\text{rank}(\Lambda_i) \approx m$ serving as a proxy (see Appendix K for discussion).

### 3.1.2 Pairwise potential enhances similarity between positive samples and prevents embeddings from collapsing

CLAMP models sub-manifolds as repulsive particles interacting via a short-range potential $E(Z_i, Z_j)$, where $Z_i, Z_j$ are the centroids of sub-manifolds $i$ and $j$. The potential attains its maximum at $||Z_i - Z_j||_2 = 0$ corresponding to complete overlap between two sub-manifolds and drops to zero when $||Z_i - Z_j||_2 > r_i + r_j$, ensuring that non-overlapping sub-manifolds do not interact. Eq. 1 implies that increasing the inter-center distance between sub-manifolds and reducing their sizes leads to a lower loss.

**Similarity.** Similarity requires that augmentations of the same image (positive samples) be mapped to nearby points in the feature space, thereby making them robust to irrelevant variations. CLAMP allows the manifold sizes to vary dynamically, unlike fixed-size particle-based models in physics. This flexibility is crucial in self-supervised representation learning, where small augmentation sub-manifold sizes $r_i$ are desirable to enhance similarity across positive samples.

**Avoiding representational collapse.** Representational collapse refers to a degenerate solution in which all network outputs converge to a constant vector, regardless of the input [31]. In CLAMP, the repulsive term in the loss function avoids such collapse by explicitly pushing sub-manifolds of different images apart.

**Separability.** Separability encourages the negative embeddings to be distant and separable on the unit hypersphere. Analogous repulsive interactions in physical systems are known to produce near-uniform distributions. Consistent with this, we observe that the embeddings progressively become more separable throughout training. To quantify this effect, we track both similarity (distance among positives) and separability (distance among negatives) throughout pretraining, confirming that CLAMP systematically enhances class-level separation.

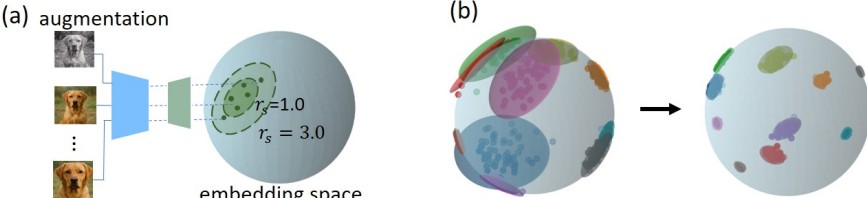

Figure 2: **Sub-manifold and visualization of the embedding space.** (a) Schematic for approximating augmentation sub-manifolds as ellipsoids with different scale factor $r_s$. (b) We selected 10 images from the MNIST dataset, one for each digit from 0 to 9, and applied Gaussian noise augmentation. These augmented images were then encoded into a 3-dimensional embedding space for visualization. Solid dots represent the embedding points of each augmented view, while the shaded regions denote circumscribed ellipsoids defined by $(\tilde{z} - Z_i)(\Lambda_i)^{-1}(\tilde{z} - Z_i) = r_s^2$. Left: the initial embeddings. Right: the trained embeddings. For this toy example, we use $r_s = 3.0$

To visualize CLAMP and provide a proof of concept, we developed a toy model using an 18-layer ResNet as the backbone, $f_\theta$, and an MLP, $g_\theta$, that maps the resulting representation to three-dimensional embedding space. We loaded one image per class from the MNIST dateset [37] and applied Gaussian noise to generate 60 augmented views per image, and set $r_s = 3.0$. Fig.2(b) shows the manifold repulsion and size shrinkage dynamics of CLAMP.

### 3.2 Implementation details

The code for CLAMP is available at `https://github.com/guanming-zhang/clamp`

#### 3.2.1 Image augmentation

During data loading, each image was transformed into $m$ augmented views using asymmetric augmentation consisting of two augmentation pipelines each sampled from different transformation distributions and applied to one half of the views. We used the following augmentations: random cropping, resizing, random horizontal flipping, random color jittering, random grayscale conversion, random Gaussian blurring (see Appendix B for details).

#### 3.2.2 Architecture

We used a ResNet network, $f_\theta(x)$, as the backbone and a MLP, $g_\phi(x)$, as the projection head. Following [1] and [2], the projection head MLP was discarded after pretraining.

**CIFAR10**    For the CIFAR10 dataset [38], we used the ResNet-18 [39] network as the backbone and a two-layer MLP of size [2048,128] as the projection head. Following [4, 2], we removed the max pooling layer and modified the first convolution layer to be `kernel_size=1,stride=1,padding=2`.

**ImageNet-100/1K**    For the ImageNet dataset [40], we used the ResNet-50 [39] network as the backbone and a three-layer MLP of size [8192,8192,512] as the projection head.

#### 3.2.3 Optimization

We used the LARS optimizer [41] for training the network. For CIFAR10, we trained the model for 1000 epochs using the warmup-decay learning rate scheduler, and with 10 warmup steps and a cosine decay for rest of the steps. For ImageNet-100, we trained the model for 200 epochs with 10 steps of warmup and 190 steps of cosine decay for the learning rate. For ImageNet-1K, we trained the model for 100 epochs with 10 steps of warmup and 90 steps of cosine decay for the learning rate. Training on 8 A100 GPUs with batch size 1024 for 4 views with distributed data parallelization for 100 epochs took approximately 17 hours. See Appendix D for details.

## 4 Evaluation

### 4.1 Linear evaluation

Following standard linear evaluation protocols, we froze the pretrained ResNet-50 backbone encoder and trained a linear classifier on top of the representation. Training was conducted for 100 epochs on ImageNet-1K and 200 epochs on ImageNet-100. Classification accuracies are reported on the corresponding validation sets (Table 1). We find that CLAMP achieves competitive performance relative to existing SSL algorithms and even sets a new state of the art on ImageNet-100. Our model performance under linear evaluation depends only weakly on batch size but improves with an increased number of augmented views as shown in Appendix L. See Appendix E for details about linear evaluation.

### 4.2 Semi-supervised learning

We evaluated the model using a semi-supervised setup with $1\%$ and $10\%$ split (the same as [1]) of the ImageNet-1K training dataset. During semi-supervised learning, both the backbone encoder and the appended linear classifier were updated for 20 epochs. The results are shown in Table 1. Again,

| Method | Linear evaluation | | Semi-supervised | |
|---|---|---|---|---|
| | ImageNet-100 | ImageNet-1K | 1% | 10% |
| SimCLR[1] | 79.64 | 66.5 | 42.6 | 61.6 |
| SwAV[29] | - | **72.1** | **49.8** | **66.9** |
| Barlow Twins[42] | 80.38* | 68.7 | 45.1 | 61.7 |
| BYOL[2] | 80.32* | 69.3 | 49.8 | 65.0 |
| VICReg[6] | 79.4 | 68.7 | 44.75 | 62.16 |
| CorInfoMax[43] | 80.48 | 69.08 | 44.89 | 64.36 |
| MoCo-V2[3] | 79.28* | 67.4 | 43.4 | 63.2 |
| SimSiam[4] | 81.6 | 68.1 | - | - |
| A&U [30] | 74.6 | 67.69 | - | - |
| MMCR (4 views+ME)[34] | 82.88 | 71.5 | 49.4 | 66.0 |
| CLAMP (4 views) | **85.12** $\pm.05$ | 69.50 $\pm.14$ | 47.38 $\pm.56$ | 65.10 $\pm.30$ |
| CLAMP (8 views) | **85.10** $\pm.15$ | 70.04 $\pm.16$ | 47.87 $\pm.03$ | 65.96 $\pm.04$ |

Table 1: **Linear evaluation accuracy on ImageNet-100 dataset for 200-epoch pretraining**: For SimCLR the higher accuracy between [34] and [43] is reported. For Barlow twins, BYOL, and MoCo-V2, the accuracies using ResNet-18 as the backbone (labeled with *) in [43] are reported because either ResNet-50 results are absent or ResNet-18 performs better in [43]. A&U and MMCR results are taken from [30] and [34] respectively. **Linear evaluation accuracy on ImageNet-1K dataset for 100-epoch pretraining**: For SimCLR, SWAV, Barlow Twins, BYOL, VICReg, CorInfoMax, MoCo-V2 and SimSiam, we reported the highest accuracy for each method among the results in the solo-learn library [44], VISSL library [45], [34] and [43]. We report the 200-epoch pretraining result for A&U, as the 100-epoch result is unavailable [30]. **Semi-supervised learning**:The semi-supervised learning results for VICReg and CorInfoMax are taken from [43], those for other methods are from [34]. In MMCR with 4 views and a momentum encoder (4 views + ME), the effective 8 views, 4 from the backbone and 4 from the encoder, achieve the same linear accuracy (71.5%) as using 8 views without the momentum encoder [34]

we find that CLAMP achieves competitive performance relative to existing SSL algorithms. See Appendix G for details.

## 4.3  Transfer to object detection tasks

To evaluate CLAMP's generalizability beyond linear classification, we apply the pretrained model to object detection. Specifically, we report results using a Faster R-CNN architecture with a C4 backbone, pretrained with CLAMP on ImageNet-1K using 8 views and a batch size of 512 for 100 epochs. The model is fine-tuned on the VOC2007+2012 training set and evaluated on the VOC2007 test set (Table. 2). CLAMP delivers stronger detection performance than other self-supervised learning baselines, demonstrating that its learned manifold structure transfers effectively to spatially structured vision tasks. See Appendix H for details.

| Methods | mAP ↑ | AP50 ↑ | AP75 ↑ |
|---|---|---|---|
| SimCLR | 54.4 | 81.6 | 61.0 |
| Barlow Twins | 53.1 | 80.9 | 57.7 |
| BYOL | 55.6 | **82.3** | 62.0 |
| MoCo v2 | 54.7 | 81.7 | 60.2 |
| MMCR | 54.6 | 81.9 | 60.0 |
| CLAMP | **55.7** | **82.3** | **62.4** |

Table 2: **Object detection on VOC datasets**: The benchmarks (mAP, AP50 and AP75) for the baseline methods are from [34]. Models are pretrained on ImageNet-1K for 100 epochs before fine-tuning.

## 5  Training dynamics reflect geometrical changes in the embedding space

We analyze manifold packing dynamics by tracking the evolution of neighbor count and manifold size during ImageNet-1K training. Two sub-manifolds are considered neighbors if the Euclidean distance between their centroids is smaller than the sum of their radii ($||q_i - q_j||2 < r_i + r_j$). At each epoch, we compute the average number of neighbors (for $m = 4$ views) and the average manifold size, $\mathbb{E}_{i\sim\text{data}}[\sqrt{\Lambda_i}/m]$, using a 1% validation split. As shown in Fig. 3, both metrics decrease over training, indicating that embeddings evolve from an initial collapsed state toward more structured representations with increased pairwise distances. This mirrors random organization models [17, 15], where local density and spacing show similar temporal behavior, reflecting improved feature discrimination over time.

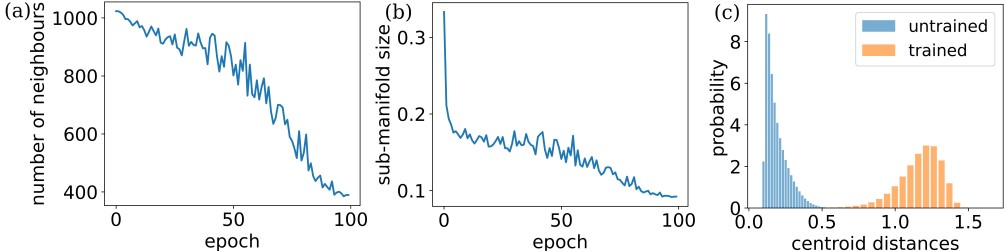

Figure 3: Training dynamics: (a) Number of neighbours as a function of epochs. (b) Average embedding sub-manifold sizes as the function of epochs. (c) Distances between pair of embeddings for untrained and trained networks.

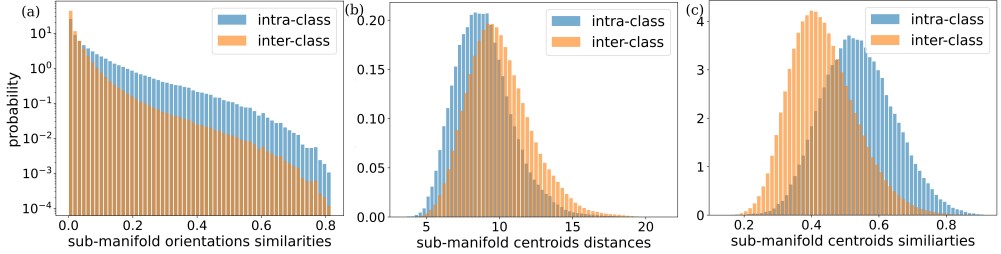

Figure 4: The properties of sub-manifolds in the embedding space for the pretrained ResNet-18 network are characterized by: (a) Orientation similarity: the squared cosine similarity between the principal orientations of the sub-manifolds. (b) Centroid distances: the Euclidean distances between the centroids of different sub-manifolds. (c) Centroid similarity: the cosine similarity between the centroid points of sub-manifolds.

## 6 Properties of the representations

To quantify the geometric properties of the learned representations, we analyzed the CLAMP representation obtained for CIFAR-10 using a ResNet-18 backbone architecture. As shown in Appendix F, we achieve $\sim 90\%$ top-1 accuracy (without hyperparameter tuning) which is competitive with alternative SSL frameworks. For each sub-manifold, we computed its centroid and alignment vector (defined as the principal eigenvector of its covariance matrix), grouping them according to their class labels. We compared the geometric properties such as centroid distances, centroid cosine similarity, and alignment cosine similarity of representations across inter-class and intra-class sub-manifold pairs. To ensure robust estimation, we randomly sampled 800 images from the test dataset and generated 100 augmentations per image to compute the manifold properties. This process was repeated 10 times.

As shown in Fig. 4, intra-class and inter-class sub-manifolds exhibit clear differences across all metrics, indicating that manifolds of different classes are distinct in

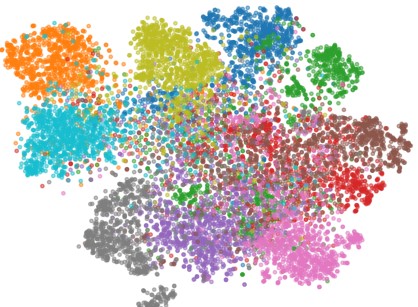

Figure 5: **t-SNE visualization of the representations.** Visualization of the 256-dimensional representation space by t-SNE method. Each color shows the representation corresponding to different category in CIFAR-10 dataset.

both position and orientation, reinforcing the potential for linear separability. Notably, the inter-class alignment similarity peaks strongly at $\sim 0$, suggesting that sub-manifolds of different classes are almost orthogonal to one another.

To demonstrate the emergence of category-specific manifolds, we visualize the representation space of the CIFAR-10 test dataset (without augmentation) using t-SNE [46], a method that preserves local neighborhood structure while projecting high-dimensional data into a lower-dimensional space. As shown in Fig. 5, representations from different classes form distinct, well-separated clusters, indicating the formation of class-specific manifolds in the learned representation space. This clear clustering underscores the structured organization induced by CLAMP and demonstrates its effectiveness in producing representation that are separable in representation space.

## 7 Biological implications

| Methods | V1 ↑ | V2 ↑ | V4 ↑ | IT ↑ |
|---|---|---|---|---|
| SimCLR | 0.224 | 0.288 | 0.576 | 0.552 |
| SwAV | 0.252 | 0.296 | 0.568 | 0.533 |
| Barlow Twins | **0.276** | 0.293 | 0.568 | 0.545 |
| BYOL | 0.274 | 0.291 | **0.585** | 0.55 |
| MMCR (8 views+ME) | 0.270 | 0.311 | 0.577 | 0.554 |
| CLAMP (4 views) | 0.258 ±0.013 | **0.336** ±0.017 | 0.558 ±0.004 | **0.570** ±0.005 |

Table 3: **Brain-score**:We use public data from [47] for V1 and V2, and from [48] for V4 and IT to evaluate the brain-score [49, 50]. The benchmarks for all baseline methods are from [34]. In the MMCR method, ME refers to the momentum encoder.

Neuroscience has long served as a source of inspiration for advancements in artificial neural networks, prompting the question of whether SSL models can replicate response statistics observed in cortex. Stringer et al. [51] analyzed stimulus-evoked activity in mouse V1 (primary visual cortex) and found that the eigenspectrum of its covariance matrix follows a power-law decay, $\lambda \propto n^{-\alpha}$ with $\alpha \approx 1.04$, and further proved that $\alpha > 1$ is indeed necessary to ensure differentiability of the neural code. After training CLAMP on ImageNet-1K, we measured the eigenspectrum of its feature covariance and observed a power law decay $\lambda \propto n^{-1.013}$, matching the cortical exponent and theoretical predictions [51].

We then evaluated representational alignment using the Brain-Score benchmark [49, 50], which measures how well model activations predict primate neural recordings via cross-validated linear regression. We found that CLAMP achieves the highest Brain-Score in V2 and IT compared to the selected self-supervised models. These results suggest that CLAMP 's representation geometry not only supports downstream classification performance

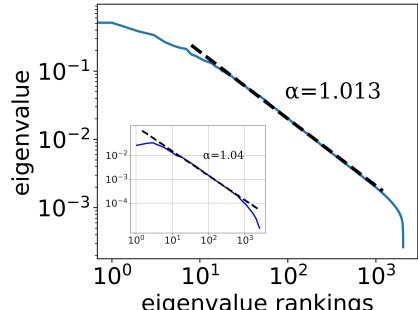

Figure 6: The eigenspectrum of the covariance matrix for the responses of 24000 images randomly selected from the ImageNet-1K dataset follows a power law decay $\lambda \propto n^{-1.013}$. Our model is pretrained on ImageNet-1K for 100 epochs. The inset is adapted from the eigenspectrum of the stimulus-evoked activity in mouse V1 reported in [51]

but also offers a compelling account of how complex visual features may self-organize in cortex without explicit supervision.

## 8 Discussion

We have introduced CLAMP, a novel self-supervised learning framework that recasts contrastive learning as a neural-manifold packing problem, guided by a physics-inspired loss. By promoting similarity of positive pairs and repulsion of negatives through purely short-range, multi-view repulsive interactions, CLAMP achieves competitive, and in some cases state-of-the-art, performance across standard downstream benchmarks. Importantly, our analysis of training dynamics reveals a progressive structuring of the embedding space. Sub-manifolds that are initially overlapping evolve into well-separated compact structures. We further show that class-level manifolds emerge from the aggregation of sub-manifolds, which interact purely through repulsion, a phenomenon reminiscent of phase separation in statistical physics. The mechanism underlying this emergent organization should be elucidated in future theoretical work. It is also interesting to test if hierarchical cluster emerges during the learning process in the future.

We demonstrate that CLAMP offers a compelling account of how complex visual features might self-organize in the cortex without explicit supervision, based on direct comparisons between its learned representations and experimental neural data. However, these findings should be interpreted with caution. Brain-Score's linear mapping between deep nets output and experimental activity may overlook non-linear relationships between model and neural representation. It also does not guarantee that CLAMP implements the same computations or dynamics as cortex. Moreover, CLAMP's use of large-batch contrastive optimization with backpropagation is unlikely to reflect biologically realistic learning mechanisms. In contrast, Hebbian-style self-supervised learning frameworks with local update rules are considered biologically plausible [52]. Developing local self-supervised learning

dynamics inspired by manifold packing, without relying on backpropagation, is an important direction for future research.

Our loss function leverages the sizes and locations of sub-manifolds within the embedding space. Incorporating orientation alignment among sub-manifolds may enable denser packing, potentially improving downstream task performance. However, we currently omit orientation information due to its high computational cost: accurately estimating a sub-manifold's orientation requires on the order of $O(D) \sim 100$ points, $\sim 100$ augmentations per input, which is infeasible at the scale of CIFAR-10 or ImageNet.

## Acknowledgments

We thank Shivang Rawat, Mathias Casiulis and Satyam Anand for valuable discussions. This work was supported by National Science Foundation grant IIS-2226387, the National Institutes of Health under award number R01MH137669, the Simons Center for Computational Physical Chemistry, and NYU IT High Performance Computing resources, services, and staff expertise. We also gratefully acknowledge the use of research computing resources provided by the Empire AI Consortium, Inc., with support from the State of New York, the Simons Foundation, and the Secunda Family Foundation.

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

# A Algorithm

---

**Algorithm 1** CLAMP algorithm

---

**Input:** batch size $b$, number of views $m$, number of optimization steps $n$,
size scaling factor $r_s$
initialize feature extractor network $f_\theta$
initialize projection head $g_\phi$
initialize the distribution of random transforms $\mathcal{T}_1$ and $\mathcal{T}_2$
**for** $t = 1$ **to** $n$ **do**
    sample a batch from the dataset $x_1, x_2, ...x_b$
    **for** $i = 1$ **to** $b$ **do**
        # for each image, augment half of the views with $\mathcal{T}_1$
        $x_i^1, ...x_i^{m/2} = t^1(x_i)..., t^{m/2}(x_i).\quad t^1...t^{m/2} \sim \mathcal{T}_1$
        #augment the other half of the views with $\mathcal{T}_2$
        $x_i^{m/2+1}, ...x_i^m = t^{m/2+1}(x_i)..., t^m(x_i).\quad t^{m/2+1}...t^m \sim \mathcal{T}_2$
        $y_i^1, ...y_i^m = f_\theta(x_i^1), ...f_\theta(x_i^m)$ # extract features
        $z_i^1, ...z_i^m = g_\phi(y_i^1), ...g_\phi(y_i^m)$ # projection
    **end for**
    # normalization
    $c = \frac{1}{mb} \sum_{i=1}^b \sum_{k=1}^m z_i^k$
    $\tilde{z}_i^k = (z_i^k - c)/\|z_i^k - c\|_2$ for $i = 1...b, k = 1...m$
    **for** $i = 1$ **to** $b$ **do**
        $q_i = \frac{1}{m} \sum_k \tilde{z}_i^k$ # center of each sub-manifold
        $\Lambda_i = \frac{1}{m} \sum_k (\tilde{z}_i^k - q_i)^T (\tilde{z}_i^k - q_i)$ # covariance matrix
        $r_i = r_s \sqrt{\frac{1}{m} \text{Tr}(\Lambda_i)}$ # manifold size
        # Note: in practice, only the diagonal elements of $\Lambda_i$ are calculated for $\text{Tr}(\Lambda_i)$.
    **end for**
    calculate the loss function
    $\mathcal{L}_{overlap} = \begin{cases} \sum_{i \neq j} \left(1 - \frac{\|q_i - q_j\|_2}{r_i + r_j}\right)^2 &,\text{if } \|q_i - q_j\|_2 < r_i + r_j \\ 0, & \text{otherwise} \end{cases}$
    $\mathcal{L} = \log(\mathcal{L}_{overlap})$
    update the network $f_\theta, g_\phi$ to minimize $\mathcal{L}$
**end for**
**return** $f_\theta$ and discard $g_\phi$

---

## B  Image augmentation

During self-supervised training, CLAMP employs a collection of image augmentations, drawn from the broader set introduced in [1] and [2]. We adopt the following image augmentations: randomly sized cropping, Gaussian blur, random gray-scale conversion(color dropping), random color jitter,randomly horizontal flipping.

| parameter | value ($\mathcal{T}_1 = \mathcal{T}_2$) |
|---|---|
| random sized cropping - output size | $32 \times 32$ |
| random sized cropping - scale | [0.08,1.0] |
| Gaussian blur - probability | 0.5 |
| Gaussian blur - kernel size | $3 \times 3$ |
| color drop - probability | 0.2 |
| color jitter - probability | 0.8 |
| color jitter - brightness adjustment max intensity | 0.8 |
| color jitter - contrast adjustment max intensity | 0.8 |
| color jitter - saturation adjustment max intensity | 0.8 |
| color jitter - hue adjustment max intensity | 0.2 |
| horizontal flipping - probability | 0.5 |

Table 4: Image augmentation parameters for CIFAR10

| parameter | $\mathcal{T}_1$ | $\mathcal{T}_2$ |
|---|---|---|
| randomly sized cropping - output size | $224 \times 224$ | $224 \times 224$ |
| randomly sized cropping - scale | [0.08,1.0] | [0.08,1.0] |
| Gaussian blur - probability | 0.8 | 0.8 |
| Gaussian blur - kernel size | $23 \times 23$ | $23 \times 23$ |
| color drop - probability | 0.2 | 0.2 |
| color jitter - probability | 0.8 | 0.8 |
| color jitter - brightness adjustment max intensity | 0.8 | 0.8 |
| color jitter - contrast adjustment max intensity | 0.8 | 0.8 |
| color jitter - saturation adjustment max intensity | 0.2 | 0.2 |
| color jitter - hue adjustment max intensity | 0.1 | 0.1 |
| horizontal flipping - probability | 0.5 | 0.5 |
| solarization - probability | 0.0 | 0.2 |

Table 5: Image augmentation parameters for ImageNet-1K/100

As the last step of the augmentation process, we apply channel-wise normalization using dataset-specific statistics. Each image channel is standardized by subtracting the dataset mean and dividing by the standard deviation. For CIFAR-10, we use (0.4914, 0.4822, 0.4465) as the mean values and (0.247,0.243,0.261) for the standard deviation. For ImageNet-100 and ImageNet-1K, the normalization parameters are (0.485, 0.456, 0.406) for the mean and (0.229, 0.224, 0.225) for the standard deviation. During resizing, we employ bicubic interpolation to preserve image quality. We use Albumentaions library for fast image augmentation [53].

## C  Time complexity for the loss function

Note that only the diagonal part of the covariance matrix $\Lambda_i$ is required in the loss function, it is not necessary to calculate the complete covariance matrix. Therefore, each iteration, the time complexity for computing sub-manifold sizes $r_i$ is $O(bmD)$, the time complexity for the pairwise loss function is $O(b^2D)$, therefore the time complexity for each iteration is $O(bD \max(m,b))$. Since the number of iteration is $N/b$ where $N$ is the size of the dataset, the complexity for calculating the loss function for one epoch is $O(ND \max(m,b))$.

## D  Self-supervised pretraining

We employ the LARS optimizer for self-supervised learning with its default trust coefficient (0.001). Our learning-rate schedule begins with a 10-step linear warm-up, after which we apply cosine decay.

To enable fully parallel training, we convert all BatchNorm layers to synchronized BatchNorm and aggregate embeddings from all GPUs when computing the loss. In addition, we exclude every bias term and all BatchNorm parameters from weight decay and use weight decay parameter $= 10^{-6}$. For CIFAR-10 dataset, we removed the max pooling layer and modified the first convolution layer to be kernel size=1,stride=1,padding=2. We use $m = 16$ views for CIFAR-10 dataset and $m = 4, 8$ for ImageNet-1K/100. $r_s = 8.5$ is applied for CIFAR-10 and ImageNet-100/1K. Other parameters are shown in Table 6 where the base learning rate = lr $\times$ batch size/256.

| dataset | backbone | MLP head | batch size | lr | momentum parameter | epochs |
|---------|----------|----------|-----------|-----|-------------------|--------|
| CIFAR-10 | ResNet-18 | [2048,256] | 128 | 2.0 | 0.9 | 1000 |
| ImageNet-100 | ResNet-50 | [8192,8192,512] | 512 | 2.0 | 0.9 | 200 |
| ImageNet-1K | ResNet-50 | [8192,8192,512] | 512 | 1.1 | 0.9 | 100 |

Table 6: Self-supervised pretraining setups for CIFAR-10 and ImageNet-1K/100

## E   Linear evaluation

We discard the projection head and freeze the backbone network during linear evaluation.

**CIFAR10 dataset:** We use the Adam optimizer with learning rate $= 0.05\times$ and batch size/ 256.0 (batch size = 1024) to train the linear classifier for 100 epochs. We apply a cosine learning rate schedule.

**ImageNet-100 dataset:** We use SGD optimizer with Nesterov momentum and weight decay to train the linear classifier where the learning rate $= 0.05\times$ batch size / 256, batch size = 1024, momentum = 0.9 and weight decay = 1e-5. During training, we transform an input image by random cropping, resizing it to $224 \times 224$, and flipping the image horizontally with probability 0.5. At test time we resize the image to $256 \times 256$ and center-crop it to a size of $224 \times 224$. We apply a cosine learning rate schedule to train the linear classifier for 100 epochs.

**ImageNet-1K dataset:** we use the same setup as ImageNet-100 but base learning rate $= 1.6\times$ batch size / 256, batch size = 1024.

## F   Linear evaluation on CIFAR10

| Method | Top1 accuracy | Top5 accuracy |
|--------|---------------|---------------|
| SimCLR | 90.74 | 99.75 |
| SwAV | 89.17 | 99.68 |
| DeepCluster V2 | 88.85 | 99.58 |
| Barlow Twins | 92.10 | 99.73 |
| BYOL | 92.58 | 99.79 |
| CLAMP (ours) | 90.21 | 99.60 |

Table 7: Linear evaluation on CIFAR10 dataset using resnet-18 as the backbone encoder. Results for Sim-CLR [1], SwAV [29], DeepCluster V2 [29],Barlow twins [42] and BYOL [2] methods are taken from solo-learn benchmarking [44].

Note that we adopt the same learning rate used for ImageNet-100 and do not perform further hyperparameter tuning.

## G   Semi-supervised learning

We use semi-supervised learning via fine tuning on the backbone network with a linear classifier for 1% and 10% splits of the ImageNet-1K dataset. The same splits are applied as [1]. We use the SGD optimizer with momentum of 0.9 but no weight decay for cross entropy loss. We adopt the same data augmentation procedure during training and testing as in the linear evaluation protocol. We scale down the learning rate of the backbone parameters by a factor of 20. Both the backbone and classifier use the cosine-decay learning rate schedule. We use a batch size of 256 and sweep the initial learning rate over {0.1, 0.2, 0.4, 0.6}.

# H    Object detection

we apply the Faster R-CNN detector with a R50-C4 backbone (pretrained on ImageNet-1K with 8 views and batch size 512 for 100 epochs), fine-tuned on VOC2007+2012 training dataset and tested on VOC2007 test dataset. Our implementation follows the procedure of [3], except that we set the base learning rate to 0.07.

# I    Power law exponent for the eigen-spectrum

In analysis of experimental data in [51], the exponent is estimated using log-log scale liner fit on eigenvalues ranked from 11 to 500. Here, we use eigenvalues ranked from 15 to 1400 to fit the exponent.

# J    Brain-Score

We use the datasets with identifiers, FreemanZiemba2013public.V1-pls, FreemanZiemba2013public.V2-pls, MajajHong2015public.V4-pls, MajajHong2015public.IT-pls for the neural activity recorded from V1,V2,V4 and IT respectively.

# K    Qualitative understanding of sub-manifolds in the embedding space

## K.1    Manifold radius for singular $\Lambda_i$

As $\Lambda_i$ is positive semi-definite, it can be decomposed as

$$\Lambda_i = \sum_j^K \lambda_i^{(j)} v_i^{(j)} v_i^{(j)^T}$$

with $\lambda_i^{(k)} > 0$ where $K = rank(\Lambda_i)$ and $v_i^{(j)}$ are orthonormal vectors. We then enclose the sub-manifold as the ellipsoid whose equation reads,

$$\frac{[(\tilde{z} - Z_i)^T v_i^{(1)}]^2}{r_s^2 \lambda_i^{(1)}} + \frac{[(\tilde{z} - Z_i)^T v_i^{(2)}]^2}{r_s^2 \lambda_i^{(2)}} + ... + \frac{[(\tilde{z} - Z_i)^T v_i^{(K)}]^2}{r_s^2 \lambda_i^{(K)}} = 1.$$

This equation represents the K-dimensional ellipsoid centered at $Z_i$ embedded in the D-dimensional space with semi-axial lengths $r_s\sqrt{\lambda_i^{(1)}}...r_s\sqrt{\lambda_i^{(K)}}$. We compute the sub-manifold radius as the square root of average semi-axial length square,

$$r_i = \sqrt{\frac{r_s^2 \lambda_i^{(1)} + ... + r_s^2 \lambda_i^{(K)}}{K}}.$$

Even if $\Lambda_i$ is not a full-rank matrix, $\text{Tr}(\Lambda_i) = \lambda_i^{(1)} + ... + \lambda_i^{(K)}$ is well defined. We get

$$r_i = r_s\sqrt{\frac{\text{Tr}(\Lambda_i)}{K}}$$

$\Lambda_i$ is the covariance matrix of $m$ embedding vectors embedded in D-dimensional space, Therefore $rank(\Lambda_i) \leq min(m, D) = m$ (in practice, the number of views is much smaller than the dimensionality , $m \ll D$). Here, we use $m$, the upper bond, to estimate $rank(\Lambda_i)$.

## K.2    The relation between estimated sub-manifold radius and the estimated sub-manifold volume

We enclose the sub-manifold $i$ by $K$ ($K = rank(\Lambda_i)$) dimensional ellipsoid embedded in the D-dimensional space with semi-axial lengths $r_s\sqrt{\lambda_i^{(1)}}...r_s\sqrt{\lambda_i^{(K)}}$. Its volume reads

$$V = \frac{\pi^{K/2}}{\Gamma(\frac{K}{2} + 1)} \prod_j^K r_s\sqrt{\lambda_i^{(j)}}.$$

From the inequality of arithmetic and geometric means, we have

$$V \leq \frac{\pi^{K/2}}{\Gamma(\frac{K}{2}+1)} \left( \frac{1}{K} \sum_i r_s \sqrt{\lambda_i^{(j)}} \right)^K$$

By Cauchy–Schwarz inequality

$$V = \frac{\pi^{K/2}}{\Gamma(\frac{K}{2}+1)} \left( \frac{1}{K} \sum_i r_s \sqrt{\lambda_i^{(j)}} \right)^K \leq \left( r_s \sqrt{\frac{\mathrm{Tr}(\Lambda_i)}{K}} \right)^K \frac{\pi^{K/2}}{\Gamma(\frac{K}{2}+1)} = \frac{\pi^{K/2}}{\Gamma(\frac{K}{2}+1)} r_i^K$$

Therefore $r_i$ is related to the upper-bound of the estimated sub-manifold volume.

## L  Hyperparameters

We conducted an ablation study to investigate the role of different hyperparameters. Due to limited computational resources, ablation study on parameter $r_s$ and number of views $m$, $30\%$ of the training data from ImageNet-1K were selected and pretrained for 100 epochs (which is consistent with our baseline where the whole training set are pretrained for 100 epochs). For the ablation on batch size, we employed the entire ImageNet-1K dataset. We used the frozen-backbone linear evaluation accuracy on the entire test dataset as the criterion for ablation.

### L.1  Scale parameter $r_s$

As shown in Fig. 7, accuracy increases with $r_s$ until it reaches a plateau at approximately $r_s \approx 6.0$. This result further supports our qualitative finding that setting $r_s > 3$ is necessary to compensate for the underestimation of manifold size. A higher $r_s$ increases computational cost because more sub-manifold pairs are involved in the loss function and the corresponding optimization process.

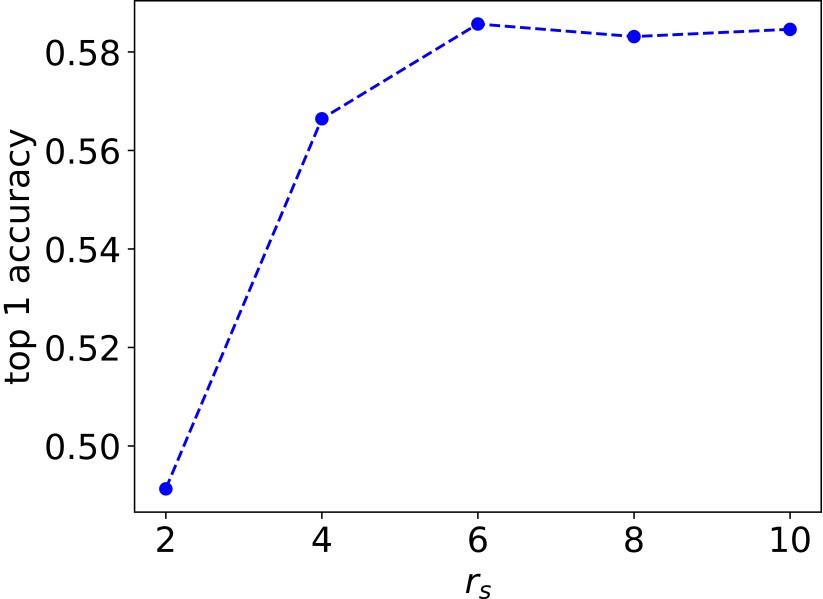

Figure 7: Linear evaluation accuracy as the function of the size scale factor $r_s$ on $30\%$ ImageNet-1K dataset. we use 4 views.

### L.2  Number of augmentations $m$

We observed that increasing the number of data augmentations leads to higher linear evaluation accuracy. This indicates that greater augmentation diversity helps the model learn better representations.

|              | 2 views | 4 views | 8 views |
|--------------|---------|---------|---------|
| top 1 accuracy | 55.17 | 58.29 | 58.95 |

Table 8: Linear evaluation using different number of views for ablation study on 30% ImageNet-1K dataset

## L.3  Batch sizes

In contrast to SimCLR, where large batch sizes are critical for achieving high accuracy, we find that the linear evaluation accuracy of CLAMP depends only weakly on batch size.

| batch size     | 256   | 512   | 1024  |
|----------------|-------|-------|-------|
| top 1 accuracy | 69.02 | 69.43 | 69.11 |

Table 9: Linear evaluation using different batch sizes for 4 views

