# OpenReview forum: "Contrastive Self-Supervised Learning As Neural Manifold Packing"
_NeurIPS.cc/2025/Conference — NeurIPS 2025 poster_

### Official Review · Reviewer_qXva · 2025-06-30

**Clarity:** 3
**Significance:** 3
**Originality:** 3
**Rating:** 5
**Confidence:** 3

**Summary:**

The paper proposes a new objective criterion for contrastive self-supervised learning. The key idea is based on: (i) a novel representation that treats different augmentations of the same image as submanifolds parameterized by ellipsoids, and (ii) a criterion that quantifies the overlap between these ellipsoids to select negative pairs. The selected ellipsoids are then used in the objective function, encouraging the network to produce similar representations within each ellipsoid while repelling the centroids of different ellipsoids. Overall, the proposed strategy can be seen as a method to effectively harness negative sample pairs.
The experimental analysis is conducted by first evaluating linear probing performance on ImageNet-100/1K, comparing the proposed method against several self-supervised baselines, and then by analyzing the properties of the learned representations along with their neuroscientific interpretations. The results demonstrate that the proposed approach performs comparably to state-of-the-art self-supervised learning methods and shows strong alignment with neuroscientific findings.

**Questions:**

Please refer to the above mentioned weaknesses.

**Ethical Concerns:**

["NO or VERY MINOR ethics concerns only"]

**Final Justification:**

Authors have clarified all weaknesses about significance and positioning with respect to existing literature. Therefore, I update my score to account for the additional information provided in the rebuttal

The work provides experimental benefits especially on the aspect regarding neuroscientific alignment and is of interest to both researchers working in the area of self-supervised learning as well as computational neuroscience.

While this is good work, I believe it is not of groundbreaking nature and therefore I wouldn't recommend it for an oral.

**Limitations:**

Yes

**Paper Formatting Concerns:**

No impact statement section is provided

**Quality:**

3

**Strengths And Weaknesses:**

**Strenghts**
- The paper is well written and easy to follow **Clarity**
- The idea of using ellipsoids and quantifying their overlap is intuitive, simple, yet original **Originality**
- The experimental methodology on linear probing evaluation is sound **Quality**.
- The experiments to evaluate the alignment with neuro-scientific findings are highly appreciated **Quality**
- Code is provided along with the submission **Code availability**

**Weaknesses**
- The main weakness lies in the significance of the proposed solution **Significance**. Since the method achieves performance comparable to the baselines in linear probing, it remains unclear whether the observations reported in Sections 5, 6, and 7 also hold for the baselines. The claims and analysis would be much stronger and more convincing if comparable results were provided for those cases.
- There is extensive literature on harnessing negative samples in contrastive learning, e.g. [1], as well as approaches that enforce clustered representations without explicit clustering, e.g. [2] **Quality**. A discussion of related work is necessary to better contextualize and highlight the novelty of the work.

**References** \
[1] Contrastive Learning with Hard Negative Samples. ICLR 2021 \
[2] Collapse-Proof Non-Contrastive Self-Supervised Learning. ICML 2025

---

> ### Author Rebuttal · Authors · 2025-07-30
>
> We thank the reviewer for the careful review. Below, we provide our detailed, point‑by‑point responses.
>
> + "The main weakness lies ..."  : In Section 5 we deliberately track sub‑manifold sizes and nearest‑neighbour counts because these metrics are intrinsic to CLAMP’s augmented submanifold loss; contrastive baselines like SimCLR, BYOL, and MoCo do not explicitly form or optimize over sub‑manifolds, so these measurements simply don’t map onto their  training dynamics.
> For Sections 6 and 7, we instead draw on the published Brain‑Score benchmarks and eigenspectrum exponents reported in the MMCR paper (NeurIPS 2023: 24103–24128). We have extracted the relevant numbers from MMCR and included them alongside our own results to enable a fair comparison, showing that CLAMP’s manifold‑packing framework yields distinct gains in neuroscientific alignment and geometric uniformity over these established methods.
>
> + "There is extensive literature ..." :  Thanks for referring to these two works. We will add the sentences below to the related work section in the finalized version.
>
>    _Hard‑negative methods draw the negative samples  from the tailored Boltzmann-like distributions to sharpen sample‑wise contrasts._
>
>   _Collapse‑proof non‑contrastive methods preserve feature diversity and avoid collapse by implicitly clustering representations and enforcing augmentation invariance without negative samples._

---

> > ### Comment · Reviewer_qXva · 2025-08-09
> > **Answer**
> >
> > Dear authors,
> >
> > thanks for the clarifications, which addressed all my concerns. I would suggest to include the whole Table 2 in [1] or report it in the Appendix so as to have a complete picture and make sure to highlight the benefit brought by the proposed strategy.
> >
> > Overall, I'm positive about the work and considering that I don't have additional suggestions to improve the paper, I'll raise my score.
> >
> > Congratulations for the work !
> >
> > [1] Efficient Coding of Natural Images using Maximum Manifold Capacity Representations. NeurIPS 2023

---

### Official Review · Reviewer_bg49 · 2025-07-03

**Clarity:** 4
**Significance:** 2
**Originality:** 4
**Rating:** 5
**Confidence:** 4

**Summary:**

A new approach for self-supervised learning of manifolds, based on the idea of packing problem - the idea that different "objects" cannot occupy the same space, so that if "overlapping" objects repel each other, we may find a packing of the objects in space in which they do not overlap. Here, "objects" are actually "object manifolds", the collections of neural responses to the same objects. If objects overlap, they cannot be classified correctly; thus, a physical-simulation-like learning where object manifolds repel each other leads to representations where objects can be reliably classified from one another.

**Questions:**

1. Do CLAMP have meta-parameters which control the balance between alignment and uniformity (lines 119-121)? i.e., control the resulting manifold uniformity despite the locality of all interactions?

**Ethical Concerns:**

["NO or VERY MINOR ethics concerns only"]

**Final Justification:**

I am satisfied with the clarifications and have adjusted my score upwards. I hope the improvements will make it to the published manuscript.

**Limitations:**

yes

**Quality:**

3

**Strengths And Weaknesses:**

## Strengths
 1. Great idea and clear presentation of the physics intuition, with our understanding of jammed objects really helping to visualise the process.
 2. The results are interesting, showing comparable performance to other methods (what the authors call "competitive").
 3. Provide a very simple framing which pushes the representation toward both uniformity (large distances between manifolds) and small size (small distances within manifolds).

## Weaknesses
 1. The "with only local operations" argument is weird and misleading: while only "near-by" objects get pushed, this push would change the representation of ALL objects (because of changes to the encoder).
 2. It is not clear which changes happen in the encoder and which happen in the embedding. As the embedding is not used for classification problems, some of the learned information is actually lost!
 3. The difference between MMCR and CLAMP is that the former consider all binary linear classifications, which CLAMP consider all binary non-linear classifications. The benefit of "local interactions" (lines 11-113) is silly as noted above.
 4. The results on brain-score are basically null-results (except for V2 maybe), showing SOME correspondence between the learned representations and the brains' representations, without a clear rule. Section 6 could have been omitted.
 5. The argument on a similar exponent like V1 is extremely weak, as there are probably 4-5 controllable parameters which affect the exponent, making the results meaningless. Section 7 could have been omitted.

---

> ### Author Rebuttal · Authors · 2025-07-30
>
> We thank the reviewer for time and efforts. Below, we provide our detailed, point‑by‑point responses.
>
> + " The "with only local operations" argument ... " : Thanks for pointing this out. The reviewer is correct that the optimization of any active pair of overlapped manifolds changes the encoder and, as a result, all representations are modified. We will clarify what we mean by "with only local operations" and will test explicitly the effect on the centroids of inactive manifolds due to an operation on the centroids of two active manifolds.
>
> + "It is not clear which ... " : Thank you for raising this point. The projection head implemented as an MLP (whose outputs we refer to as embeddings) acts as an effective scaffold during contrastive training. In both SimCLR (arXiv:2002.05709) and MoCo v2 (arXiv:2003.04297), inserting this projection head during training and removing it at inference time consistently improves downstream classification accuracy compared to using no projection head at all. Following their success, subsequent methods such as BYOL and Barlow Twins have adopted the same architecture. Here, we also incorporate an MLP projection head to enhance our model’s training performance. In our revised ablation study, we will test the effect of not including the projection head.
>
> + "The difference between MMCR ..." : MMCR seeks to maximize the “manifold capacity” of a set of N low dimensional manifolds. This should be understood in the sense of “perceptron capacity”, i.e. MMCR seeks to improve the relative arrangement of the manifolds so that a linear classifier can shatter as many random assignments of binary labels (1 or 0) to the N manifolds, in certain technical limits. So MMCR ignores the fact that the N manifolds belong to multiple distinct classes, and instead probes the problem as a linear binary classification of low-dimensional manifolds carrying one of two labels. So, MMCR is not explicitly trying to pack manifolds, but it achieves manifold separation because it tends to favor higher perceptron capacity (note however that this is quite indirect and less interpretable than the clear physical picture provided by CLAMP). CLAMP instead directly addresses the N-ary problem of packing and nonlinearly classifying N distinct manifolds in a simple and physically interpretable framework. We will clarify what we mean by local interactions as stated above, and will test explicitly the effect on the centroids of inactive manifolds due to an operation on the centroids of two active manifolds.
>
> + " The results on brain-score ..." :  We appreciate the reviewer’s suggestions about our Brain‑Score (Section 6). Our intent in these sections was not to claim that CLAMP perfectly mirrors every stage of visual cortex, but rather to probe whether a repulsive objective, with no explicit biological constraints, nonetheless gives rise to any of the statistical hallmarks we observe in real neural populations and shows the potential of implementing CLAMP by biologically plausible learning rules.  The limitations of Brain-Score have been discussed in section 8.  We will modify section 8 to show more detailed limitations about the Brain-Score.
>
> _Brain-Score’s linear mapping between deep nets output and experimental activity may overlook non-linear relationships between model and neural representation. It also does not  guarantee that CLAMP implements the same computations or dynamics as cortex._
>
> + "The argument on a similar ..." : We appreciate the reviewer’s feedback on Section 7. As noted by Stringer et al. (Nature, 571(7765):361–365, 2019), smooth neural manifolds are expected to exhibit a power-law exponent greater than 1 across a range of experiments involving different input types and varying fractions of sampled neurons. In this context, our analysis demonstrates that the representations learned by CLAMP exhibit a similar power-law behavior. While we agree that multiple controllable parameters can influence the exponent, we view our findings as a meaningful extension of Stringer et al.’s test, providing an additional perspective on the structure of learned representations.
>
> + "Do CLAMP have meta-parameters ..." :  Thanks for the question. CLAMP’s packing loss is self-balancing: by minimizing the objective it either shrinks submanifold sizes (improving alignment) or pushes them farther apart (improving uniformity), with no extra meta-parameter required to trade off these effects. In other words, the alignment–uniformity balance emerges implicitly from the packing loss itself. As we demonstrate in Section 4.1:Linear Evaluation, the CLAMP packing loss not only simplifies the formulation but also outperforms the original alignment-uniformity method.

---

> > ### Comment · Reviewer_bg49 · 2025-08-05
> > **Response to Authors**
> >
> > Thank you for your answers. I am satisfied with them and will adjust my score upwards.

---

### Official Review · Reviewer_4tLf · 2025-07-04

**Clarity:** 3
**Significance:** 3
**Originality:** 3
**Rating:** 5
**Confidence:** 2

**Summary:**

The paper presents Contrastive Learning As Manifold Packing (CLAMP), a new self-supervised learning method for visual representations. The key idea is to model the embeddings from augmented views of an image as a "sub-manifold," which is then approximated by a high-dimensional ellipsoid. Learning is driven by a single loss function inspired by short-range repulsive potentials in physical systems. This loss function simultaneously encourages the separation of sub-manifolds from different images (negative pairs) and the contraction of individual sub-manifold volumes (positive pairs). The authors argue this unified approach is elegant, interpretable, and effective at preventing representational collapse. The method is evaluated on CIFAR-10, ImageNet-100, and ImageNet-1K using linear probing and semi-supervised learning protocols , where it demonstrates promising, and in one case, state-of-the-art results.

**Questions:**

1.	Given that the central claim is about learning better-structured representations through geometric packing, wouldn't evaluation on tasks that heavily rely on rich spatial features (e.g., object detection, segmentation) be the most convincing way to validate this?
2.	The paper's claims hinge on the ellipsoid approximation being a valid model for augmentation sub-manifolds. What empirical evidence can the authors provide to show this is a robust choice? For instance, how do the results change if a more complex geometric model is used, or when faced with augmentations that might produce a non-convex or multi-modal distribution of embeddings?
3.	In the training dynamics (Figure 3b), the sub-manifold size decreases as expected. What mechanism in the framework prevents a trivial solution where the radii collapse to near-zero values for all sub-manifolds, thus minimizing the loss?
4.	The performance of many contrastive methods is highly dependent on batch size. Since CLAMP's loss is computed over all pairs in a batch, it likely shares this dependency. How does CLAMP's performance scale with varying batch sizes?

**Ethical Concerns:**

["NO or VERY MINOR ethics concerns only"]

**Final Justification:**

The paper presents a novel self-supervised learning method of Contrastive Learning As Manifold Packing (CLAMP) for visual representations. In the rebuttal, the new object detection results have effectively addressed my main concerns regarding the evaluation on downstream tasks.

**Limitations:**

1.	The primary limitation is the scope of the evaluation. As detailed in the weaknesses, the lack of experiments on diverse downstream tasks makes it difficult to fully assess the quality of the representations learned by CLAMP. This should be explicitly stated as a key limitation of the current work.
2.	The simplification of manifold geometry by ignoring orientation and using an isotropic radius approximation is a significant constraint, as the authors themselves note. This limits the "packing" from achieving its full theoretical potential and makes the physics analogy less direct than it could be.
3.	The framework's effectiveness is intrinsically tied to the choice of data augmentations, as these define the very geometry of the sub-manifolds being packed. The paper would benefit from a more thorough discussion of this dependency and its implications.

**Quality:**

3

**Strengths And Weaknesses:**

Strengths
1.	The main contribution is the novel framing of SSL as a manifold packing problem. This is a refreshing and intuitive departure from point-wise contrastive methods.
2.	The physics-based motivation provides a clear mental model, and the resulting loss function (Eq. 1) is elegant. Its hyperparameters, like the scaling factor have a tangible physical meaning, which is a commendable feature.
Weaknesses
1.	This is the most significant weakness. The paper's empirical support is not comprehensive enough to demonstrate the superiority or generalizability of the proposed method.
2.	The evaluation is restricted to linear and semi-supervised classification. Modern SSL methods are expected to be benchmarked on a wider array of downstream tasks, such as object detection and semantic segmentation on datasets like COCO, to prove the robustness and richness of the learned features. Without this, it is difficult to ascertain if CLAMP's representations are truly general-purpose.
3.	While ImageNet and CIFAR-10 are foundational, the SSL field is moving towards more complex and varied datasets. Demonstrating strong performance exclusively on these benchmarks is no longer a sufficient condition for claiming SOTA status for a general vision backbone.
4.	The ablation study for key hyperparameters (Appendix K) was performed on a 30% subset of the ImageNet-1K training data. While the authors cite computational constraints, this reduces the reliability of the findings and makes it difficult to assess the true sensitivity of the model to parameters like and the number of views m.
5.	The framework is built on strong simplifying assumptions whose validity is not sufficiently proven by the limited experiments. The approximation of a complex augmentation distribution as a single ellipsoid is a major simplification. The paper does not provide empirical analysis to justify that this geometric approximation is robust across different augmentation strategies or that it captures the essential structure needed for learning. The limited experiments do not provide enough evidence to bridge the gap between the elegant theory and the practical implementation.

---

> ### Author Rebuttal · Authors · 2025-07-30
>
> We thank the reviewer for the constructive feedback. Below, we provide our detailed, point‑by‑point responses.
>
> + "Given that the central ..." :  we present preliminary results on object detection using a Faster R-CNN head with a C4 backbone (pretrained on ImageNet-1K with 4 views for 100 epochs), fine-tuned on VOC2007+2012 training dataset and tested on VOC2007 test dataset. Comparative metrics for other methods are sourced from the MMCR paper (arXiv:2303.03307).
> | VOC2007+12| mAP | AP50 | AP75|
> | --------------- | :-------------: | ---------------:|---------------:|
> | Barlow Twins   | 53.1  | 80.9   | 57.7   |
> | SimCLR           | 54.4  | 81.6   | 61.0   |
> | MoCo v2          | 54.7  | 81.7   | 60.2   |
> | BYOL               | 55.6  | 82.3   | 62.0   |
> | MMCR             | 54.6  | 81.9   | 60.0   |
> | CLAMP            | 55.6  | 82.2   | 61.5   |
> *Higher mAP/ AP50/AP75 shows better performance.
>   Overall our results show that CLAMP is competitive with the state of the art. We will further improve object detection by training on the larger COCO dataset and through hyperparameter tuning. We will include the new results in the revised version of the manuscript.
>
> + "The paper's claims hinge ..." : For perturbative Gaussian noise as the augmentation transformation where the input is x and the transformation is $x+\zeta$  with $\zeta \sim \mathcal{N}(0, \Sigma)$, the augmentation submanifold is provably elliptical. The resulting augmented embedding follows a Gaussian distribution:
> $ \mathcal{N}\left(g(f(x)), J_x(g(f(x))) \  \Sigma \ J_x(g(f(x)))^\top \right) + O(\zeta ^3)$,
> where $J_x( \cdot)$​ denotes the Jacobian matix. $f(\cdot)$ and $g(\cdot)$ denote the encoder and projector network.
> For more general augmentations, the resulting submanifold is not necessarily elliptical. In such cases, we can approximate the augmentation submanifold with a sufficiently large enclosing sphere, as done in CLAMP. However, this approximation may lose fine geometric details and underutilize the space’s packing capability. Conversely, modeling the submanifold with more accurate geometries  demands significantly more augmentations to estimate, on the order of $O(D)∼100-1000 $ in high-dimensional spaces, which is computationally costly untenable at scale. Thus, there is a trade-off between augmentation cost and packing efficiency.We have mentioned this point in Section 8, line 351.
>
> + "In the training dynamics (Figure 3b) ..." :  A zero-volume submanifold occurs when an image and all its augmentations collapse to the same representation, reflecting perfect transformation invariance. While this invariance can aid image classification, it may discard the fine details needed for other downstream tasks. CLAMP’s packing loss naturally prevents such collapse: radii only contract while two submanifolds overlap, and once they become merely tangent, the shrinkage gradient disappears. As a result, each radius settles at a nonzero “margin” value instead of vanishing.
>
> + "The performance of many ... " : Thank you for raising this point. Here we show the linear evaluation results of pretrained models on ImageNet-1K for 100 epochs with n_views=4 and rs=8.5.
> | Batch sizes | 256 | 512 | 1024 |
> | --------------- | :-------------: | ---------------:|---------------:|
> | Linear evaluation   | 69.0  | 69.4   | 69.1   |
>
> The training performance depends weakly on the batch size. Such weak dependency   has also been observed for MMCR.We will further improve the ablation study by including this table and additional results that we are currently generating for different numbers of views.

---

> > ### Comment · Reviewer_4tLf · 2025-08-06
> >
> > Thank you for the detailed rebuttal. The new object detection results have effectively addressed my main concerns regarding the evaluation on downstream tasks. Given the thorough response, I will raise my score.

---

### Official Review · Reviewer_AN7f · 2025-07-08

**Clarity:** 3
**Significance:** 3
**Originality:** 3
**Rating:** 5
**Confidence:** 3

**Summary:**

This paper proposes a novel reframing of contrastive self-supervised learning (CSSL) by casting it as a manifold packing problem in representation space. Instead of viewing representation learning as a purely statistical or mutual information maximization task, the authors approach it geometrically: they introduce a new loss function inspired by the short-range repulsive forces in particle physics, encouraging the learned representations to compactly fill space with minimal overlap. The paper offers an elegant and theoretically grounded framework for understanding representation learning through the lens of neural manifold packing. The fact that it achieves the best brain score for V2 and IT suggests a potential computational-level convergence between this framework and the primate visual systems.

**Questions:**

1. Does manifold packing scale to hierarchical or compositional concepts?
2. What practical benefits does this new perspective offer for self-supervised learning?  Improvement on the loss function? Does it lead to the design of new objective?

**Ethical Concerns:**

["NO or VERY MINOR ethics concerns only"]

**Final Justification:**

Thank you for the responses, clarification, and the new results concerning the practical utility and application values. I do think it is an interesting paper. I guess extending to a compositional hierarchical system might be too hard a question for you to answer at this point, and not critical for this paper.  I will keep the current favorable score.

**Limitations:**

1. While similar geometric structuring of representations (low-dimensional manifolds, separation of class-specific activity) has been observed in neural population codes in primate cortex, it is not certain how this can be implemented using biological mechanisms. Is this primarily a computational theory level description of the principle?
2. It is not certain how manifold the packing framework scales to hierarchical or compositional concepts, or large dataset.

**Paper Formatting Concerns:**

No issue.

**Quality:**

3

**Strengths And Weaknesses:**

Strengths:
1. A major contribution is the theoretical and empirical demonstration that this formulation leads to the emergence of low-dimensional, well-separated manifolds, each corresponding to different semantic image categories—even in the absence of labels. This structure supports linear decodability and provides a geometric rationale for the effectiveness of self-supervised representations.
2. Experiments on image datasets (CIFAR-10, STL-10, etc.) confirm that learned representations exhibit this packing behavior. Even without labels, the manifolds align with semantic classes.
3. One of the most striking results is that the learned representations achieve state-of-the-art brain scores in both area V2 and IT cortex when evaluated against neural data, suggesting a degree of alignment between the structure of artificial representations and those found in the primate visual system. Similar geometric structuring of representations (low-dimensional manifolds, separation of class-specific activity) has been observed in neural population codes in primate cortex. While the biological plausibility of the learning mechanisms remains uncertain, the computational-level similarity (in Marr's terms) offers intriguing evidence that efficient manifold packing may be a shared organizational principle.
4. The paper brings much-needed theoretical clarity to the success of contrastive learning. It contributes not only a practical improvement in self-supervised learning but also a conceptual bridge between deep learning and neuroscience. It provides a unifying geometric perspective that connects contrastive objectives, biological representations, and manifold theory.

Weaknesses:
1. While the geometric structure aligns well with cortical representations, the underlying mechanisms—such as particle-inspired repulsion—are not themselves biologically grounded, and more work is needed to connect this theory to known neurophysiological learning rules.
2. Evaluation: The evaluation presumes alignment between data structure and downstream labels. It's less clear how CLAMP behaves when class structure is ambiguous or hierarchical.
3. While packing helps linear separability, how does it generalize to unseen concepts or transfer across tasks?
4. The practical benefit is modest. The experiments are solid but limited to standard benchmarks; it would be valuable to see if the same geometric structure emerges in larger-scale vision-language or multimodal settings.

---

> ### Author Rebuttal · Authors · 2025-07-30
>
> We thank the reviewer for the time and efforts on reviewing our work, which have helped improve our manuscript. Below, please find our detailed responses.
> + " While the geometric structure..." :  CLAMP’s repulsive packing loss isn’t directly grounded on a specific synaptic learning rule. However, its main idea, encouraging competition and decorrelation among sub‑manifolds may be analogous to anti‑Hebbian plasticity. Bridging CLAMP’s high‑level objective with a concrete, biologically‑plausible learning model remains a very important direction for future work. We have mentioned this point in Section 8, line 343-346.
>
> + "Evaluation: The evaluation ..." : Thanks for raising this interesting point. Our t‑SNE visualization reveals that categorical labels naturally segregate suggesting an emergent structure in the embedding space. By analogy, we hypothesize that CLAMP’s representations will likewise organize into hierarchical clusters (subcategorical clusters within parent categories). We will discuss this hypothesis and seek supporting evidence for it on ImageNet in the revised version of the manuscript.
>
>
> + " While packing helps ..." : The representation learned by CLAMP demonstrates its ability to distinguish between different categories, indicating strong classification capabilities. This property is beneficial for a variety of downstream tasks. Accordingly, fine-tuning self-supervised models on object detection and segmentation tasks often yields high performance. Here, we present preliminary results on object detection using a Faster R-CNN head with a C4 backbone (pretrained on ImageNet-1K with 4 views for 100 epochs), fine-tuned on VOC2007+2012 training dataset and tested on VOC2007 test dataset. Comparative metrics for other methods are sourced from the MMCR paper (arXiv:2303.03307).
> | VOC2007+12| mAP | AP50 | AP75|
> | --------------- | :-------------: | ---------------:|---------------:|
> | Barlow Twins   | 53.1  | 80.9   | 57.7   |
> | SimCLR           | 54.4  | 81.6   | 61.0   |
> | MoCo v2          | 54.7  | 81.7   | 60.2   |
> | BYOL               | 55.6  | 82.3   | 62.0   |
> | MMCR             | 54.6  | 81.9   | 60.0   |
> | CLAMP            | 55.6  | 82.2   | 61.5   |
> *Higher mAP/ AP50/AP75 shows better performance.
>  Overall our results show that CLAMP is competitive with the state of the art. We will further improve object detection by training on the larger COCO dataset and through hyperparameter tuning. We will include the new results in the revised version of the manuscript.
>
> + "The practical benefit ... " : It is interesting to apply this type of self-supervised loss function to other tasks including multimodal tasks. We will mention this possibility in the discussion and reserve a detailed benchmark on these large-scale datasets for future work, commensurably with the available computational resources.
>
> + "What practical benefits ..." : CLAMP uses a single repulsive potential whose parameters  have clear geometric meaning, making the training dynamics tractable. The manifold‑packing view immediately suggests richer objectives such as adaptive radius schedules analogous to the Lubachevsky-Stilliger packing algorithm. We will include this discussion on the possibility of deriving new physically-motivated objectives in Section 8.

---

> > ### Comment · Reviewer_AN7f · 2025-08-04
> >
> > Thank you for the responses, clarification, and the new results.  I do think it is an interesting paper. I will keep the current favorable score.

---

### Note · Authors · 2025-08-15

We thank the reviewers, AC, and SAC for their thoughtful reviews and discussions. CLAMP reframes self-supervised learning as manifold packing with a single, interpretable short-range potential as the loss function. This physics-inspired objective jointly shrinks sub-manifolds and increases their separation, with geometrically meaningful hyperparameters. It yields strong results and interpretable embedding-space dynamics that parallel jamming physics. Empirically, CLAMP delivers state-of-the-art linear-probe accuracy on ImageNet-100, competitive ImageNet-1K performance with 4 views, and strong semi-supervised results. Beyond accuracy, we analyze the geometry and neuroscience alignment of the learned representations: sub-manifold radii shrink and the number of neighbors decreases; embeddings show clear class separation; the representation eigenspectrum follows a power law consistent with cortical experiments; and Brain-Score comparisons highlight potential parallels between learned and neural representations.

In response to the feedback, we added preliminary VOC detection results, clarified why the loss prevents collapse (radii stabilize at non-zero margins once manifolds are tangent), and showed weak batch-size dependence. We also clarified terminology (e.g., “local interactions”), expanded related work (hard negatives; collapse-proof SSL), and deepened the discussion (geometry approximation; Brain-Score considerations). These edits are incorporated in the revised manuscript. We are grateful for the constructive feedback, which further improves the work.

Finally, CLAMP is easy to adopt and flexible enough to support new objectives (e.g., adaptive radius learning schedules, applications to hierarchical data). Because the loss acts directly on the representation geometry, it can be naturally transferred to other domains and may inspire biologically plausible learning rules grounded in manifold-packing dynamics.

---

### Decision · Program_Chairs · 2025-09-17

**Decision:**

Accept (poster)

**Comment:**

The paper explores self-supervised contrastive learning through the lens of manifold geometry, drawing inspiration from neuroscience and physics. Building on the observation that neuronal responses in the cortex form structured manifolds, the authors propose Contrastive Learning As Manifold Packing (CLAMP), a framework that reformulates representation learning as a packing problem in embedding space. The key idea is to treat augmented views of an image as sub-manifolds, whose positions and sizes are optimized using a loss function inspired by the repulsive interactions in particle systems studied in condensed matter physics. This “packing loss” provides geometrically interpretable dynamics and introduces hyperparameters tied to manifold separation. The approach offers an interpretable physical analogy to jamming and packing phenomena. The authors also show the practical effectiveness, achieving competitive results under the linear evaluation protocol relative to state-of-the-art contrastive methods.

Reviewers consistently praised the conceptual novelty of linking representation learning with packing dynamics, the clarity of the geometric intuition, and the interpretability of the loss function, whose hyperparameters have clear meaning. Empirical strengths include competitive linear evaluation results on ImageNet and CIFAR benchmarks, neuroscientific alignment evidenced by strong Brain-Score performance, and new results on object detection that help validate downstream utility. The reviewers also appreciated that the work bridges machine learning with insights from neuroscience and physics. Some weaknesses that the reviewers noted was the method’s biological grounding is more metaphorical than mechanistic, the evaluation was initially limited to standard classification tasks, and the ellipsoid approximation of augmentation sub-manifolds may oversimplify real geometric structure. Some reviewers also highlighted that while the framework is elegant, the practical gains are modest compared to existing SOTA approaches, and its robustness on larger-scale or multimodal tasks remains to be demonstrated. The rebuttal seems to have addressed these concerns with new experimental results, clarifications on collapse avoidance and batch-size sensitivity, and a discussion of broader applicability. The reviewers all recommend acceptance and I concur.